# Novel Fusidic Acid Cream Containing Metal Ions and Natural Products against Multidrug-Resistant Bacteria

**DOI:** 10.3390/pharmaceutics14081638

**Published:** 2022-08-05

**Authors:** Hani Naseef, Yousef Sahoury, Mohammad Farraj, Moammal Qurt, Abdallah D. Abukhalil, Nidal Jaradat, Israr Sabri, Abdullah K. Rabba, Mahmmoud Sbeih

**Affiliations:** 1Pharmacy Department, Faculty of Pharmacy, Nursing and Health Professions, Birzeit University, Ramallah P.O. Box 14, Palestine; 2Master Program in Clinical Laboratory Science, Faculty of Pharmacy, Nursing and Health Professions, Birzeit University, Ramallah P.O. Box 14, Palestine; 3Department of Pharmacy, Faculty of Medicine and Health Sciences, An-Najah National University, Nablus P.O. Box 7, Palestine; 4Quality Control Department, Beit-Jala Pharmaceutical Co., Ltd., Bethlehem P.O. Box 58, Palestine

**Keywords:** fusidic acid, formulation, MRSA, natural products, skin infections, FRSA, antimicrobial activity

## Abstract

Background: Drug design and development to overcome antimicrobial resistance continues to be an area of research due to the evolution of microbial resistance mechanisms and the necessity for new treatments. Natural products have been used since the dawn of medicine to heal skin infections. The antimicrobial properties of fusidic acid, zinc sulfate, and copper sulfate have been studied and are well known. Furthermore, these compounds have different mechanisms of action in targeting microorganisms, either by inhibiting protein synthesis or bacterial cell walls. Therefore, their combination is expected to have synergistic activity in killing bacteria. However, the synergistic antimicrobial activity has not been evaluated in a cream formulation. Therefore, the objectives of this in vitro study were to develop and evaluate the synergistic efficacy of fusidic acid in combinations with natural products, including oleuropein, thyme oil, zinc sulfate, and copper sulfate, as a cream to eradicate fusidic-acid-resistant microorganisms in skin infections. Methods: Three different cream formulations were developed, compared, and labeled F1, F2, and F3. The compounds were studied for their antibacterial activity. In addition, the stability of the cream was investigated at 25 °C and 40 °C in plastic jars over three months. Results: The F2 formula has adequate physicochemical properties. Furthermore, it displays stable and better results than the marketed trade product and has potential inhibition zones (ZOI). Interestingly, considerable numbers (9.5%) of fusidic-acid-resistant Staphylococcus aureus (FRSA) isolates possessed a high resistance pattern with MIC ≥ 128 μg/mL. In contrast, most tested FRSA isolates (90.5%) had a low resistance pattern with MIC ≤ 8 μg/mL. Conclusion: In conclusion, the F2 cream made with fusidic acid, oleuropein, thyme oil, zinc sulfate, and copper sulfate in the right amounts has stable physical and chemical properties and has potential against FRSA as an antimicrobial agent.

## 1. Introduction

Creams are the most commonly used pharmaceutical preparations available for dermatological uses [1]. Creams have been described as semi-solid emulsions or emulsions of elevated viscosity produced for topical application [2,3,4]. The recent classification and definition of creams define them as semi-solid dosage forms consisting of more than 20% water or volatile components (hydrocarbons) and about 50% wax or polyethylene glycol as a vehicle for exterior skin application [5]. Pharmaceutical creams may contain one or more active pharmaceutical ingredients (APIs) dispersed in oil in water (O/W) or water in oil (W/O) systems. Different physicochemical destabilizing processes that depend on time and temperature have been used to describe the physical instability of creams. These processes are coalescence, creaming, flocculation, phase inversion, sedimentation, and Ostwald ripening [6].

Fusidic acid is an effective antibiotic used to treat many infections caused by methicillin-resistant Staphylococcus aureus (MRSA) and methicillin-susceptible Staphylococcus aureus (MSSA) [7]. Fusidic acid has a narrow spectrum of bacteriostatic antibiotic activity and has been used clinically since the 1960s [8]. Fusidic acid inhibits bacterial protein synthesis by binding to elongation factor G (EF-G) on the ribosomes and, therefore, prevents nascent polypeptides’ elongation [9]. Fusidic acid resistance in *S. aureus* usually develops by altering or protecting the drug target; these mechanisms have different underlying molecular mechanisms [10]. Reports from various geographical areas worldwide have revealed that the prevalence of resistance of *S. aureus* strains to fusidic acid has remained low in recent years [11,12,13]. Natural products and several ions have been used to overcome multidrug resistance in bacteria [14,15,16].

Copper sulfate pentahydrate (CuSO_4_·5H_2_O) and zinc sulfate heptahydrate USP (ZnSO_4_·7H_2_O) are used in many topical formulations owing to their antibacterial and antiviral activities [17]. Furthermore, evidence suggests that copper can inhibit some pathogens, including MRSA, influenza virus, Pseudomonas species [18,19,20], and *E. coli* [21]. The proposed mechanism of action of copper sulfate includes binding to protein molecules and denaturation of DNA; furthermore, it produces free radicals that damage the cell integrity and inhibit enzymes necessary for the vital functions of bacteria [22,23].

Zinc has been used as a single ingredient in topical products or in combination with other components due to its protective skin effects against inflammation and dermatological infections. These activities are due to the inhibition of macrophages, neutrophils, and inflammatory cytokines [24]. In addition, zinc is necessary for microbial cellular construction, metabolism, DNA and protein synthesis, and wound healing [25].

The possibility of the synergistic effects of zinc and copper against *E. coli* and *S. aureus* has been evaluated and showed significant activity at different concentrations. The 3% combination of zinc and copper has superior antimicrobial activity on the tested organisms [26,27]. However, these ions’ high reactivity and incorporation often create a formulation challenge. In addition, the oxidation reaction catalysis leads to some changes, including changes in efficacy, physical and/or chemical instability, the shelf-life of formulations, odor formation, and color and quality change [28]. Therefore, the stability of the product and the antibacterial activity were studied for 12 weeks at two different temperatures.

In Palestine, there is a lack of studies regarding the prevalence of fusidic acid resistance among clinical isolates of *S. aureus* (FRSA). However, a few reports indicate increased fusidic acid use, particularly in recurrent topical infections caused by *S. aureus*. Therefore, the current investigation aims to develop and evaluate the synergistic efficacy of fusidic acid in combinations with natural products, including oleuropein, thyme oil, zinc sulfate, and copper sulfate, in a cream formulation to eradicate fusidic-acid-resistant microorganisms in skin infections.

## 2. Materials and Methods

### 2.1. Materials

All materials used in this research met the purity requirements for pharmaceutical and analytical grades. The materials were donated by Beit-Jala Pharmaceutical Co., Ltd. Bethlehem, Palestine, including fusidic acid, zinc sulfate, copper sulfate, oleuropein, thyme oil, cetostearyl alcohol, macrogol, A6 macrogol A25, paraffin oil, propylene glycol, and Fucidin TM.

### 2.2. Methods

#### 2.2.1. Formulation of the Topical Cream

The cream base excipients were selected based on the marketed product Zydex TM (Beit-Jala Pharmaceutical Co., Ltd., Bethlehem, Palestine). The formulations were prepared using a hot emulsification method, where the oily and aqueous phases were melted separately at (60 °C and 70 °C, respectively) and then mixed. The oil phase consists of cetostearyl alcohol, macrogol A6 and A25, thyme oil, and paraffin oil with dispersed oleuropein. The concentrations of thyme oil and oleuropein were determined based on a previous study carried out in our lab and published elsewhere [29]; these substances showed effectiveness against selected microorganisms [29]. The aqueous phase consists of purified water, fusidic acid, zinc sulfate, copper sulfate, and propylene glycol. The final formulation was prepared as follows: the oil phase was slowly added to the aqueous phase and mixed with a vertical mixer for 5 min, followed by stirring using a stirrer until reaching a temperature of 40 °C. After extensive experimental trials, three formulations (F1, F2, F3) of 500 g were prepared. The composition of the formulations is shown in Table 1. All formulations were placed in plastic containers and stored for further evaluation.

#### 2.2.2. In Vitro Antibacterial Activity

The antimicrobial activity of the formulas was determined using the agar diffusion method, and the zoon inhibition diameter was measured. Each prepared formula was tested in triplicate. The microorganisms tested (*E. coli* ATCC 8739, *S. aureus* ATCC 6538, FRSA) were cultured on sterile tryptic soy agar plates. The inoculum suspension equivalent to a 0.5 McFarland standard was prepared from an overnight culture and then diluted 1:100 in MH broth according to the guidelines of the Clinical Laboratory Standards Institute (CLSI) [30]. An aliquot of 100 µL was placed on the surface of MH agar plates and then spread by a sterile glass rod. The plates were then left for 15 min to absorb the inoculum. A sterile cork borer was used to make wells of 5 mm in diameter by punching the inoculated MH agar plates. The cut agar was removed from the wells using a sterile needle. An aliquot of 0.2 mL of the samples and their dilutions was incubated at 37 °C for three days. The formulation with the minor concentration of excipients that gave the best antibacterial effects was selected as the optimal formulation for further testing in the topical cream.

### 2.3. Chemical and Physical Evaluation of the Optimal Formulation (F2)

#### 2.3.1. Determination of Oleuropein by HPLC

Phosphate buffer (pH 3.0) was prepared by mixing equal volumes of 0.01M phosphoric acid and 0.01 M monobasic sodium phosphate. The mobile phase was prepared by filtering and degassing a mixture (20:80) of acetonitrile and phosphate buffer (pH 3.0). The mobile phase also contains 1% acetic acid. The test solution was prepared by mixing 70 mg of oleuropein dissolved in a mixture of acetonitrile and water (20:80) in 100 mL volumetric flasks [31]. High-performance liquid chromatography (HPLC) equipped with a 280 nm detector and silica-based C18 bonded phase column C18, 5 μm (5 μm, 150 × 4.6 mm inner diameter) was used. The flow rate was 1.0 mL/min, and the injection volume was 20.0 μL.

#### 2.3.2. Determination of Fusidic Acid by HPLC

Phosphate buffer (pH 3.5) was prepared by mixing equal volumes of 0.01 M phosphoric acid and 0.01 M monobasic sodium phosphate. The mobile phase was prepared by filtering and degassing a mixture of methanol, 1.0 g potassium dihydrogen phosphate dissolved in 500 mL purified water, and a mixture of phosphoric acid with acetonitrile pH 3.5 (20:45:35). The test solution was prepared by dissolving 40 mg of fusidic acid in a diluent of ethanol and purified water (70:30) in a 200 mL volumetric flask. This process was performed using high-performance liquid chromatography (HPLC) equipped with a 235 nm detector and Lichrosphere RP-select B, 5μm (125 × 4 mm). The flow rate was 1.5 mL/min, and the injection volume was (20 μL).

#### 2.3.3. Determination of Zinc Sulfate Heptahydrate and Copper Sulfate Pentahydrate

Determination of the assay for (ZnSO4·7H2O) and (CuSO4·5H2O) was prepared by a complex metric titration method because of the unavailability of ICP-MS for the analysis of trace elements. The percent content was calculated by the equation: ions % = TS × C1 × C2 × 100/C3, where: TS: titration consumption in ml, C1: corresponds to ions in mg/mL (0.1 mol/L × molar mass g/mol) C2: titer EDTA (dimensionless unit), C3: sample weight in mg. The titration was conducted with 0.1 M EDTA to achieve the green endpoint at accelerated conditions (40 ± 2 °C/75% ± 5% RH) [32]. Once the green endpoint was achieved, a half a milligram (0.5 mg) of the F2 formula was mixed with 100 mL purified water, 1 mL of concentrated acetic acid, 2.5 mg of xylenol orange, and a sufficient amount of a hexamine mixture ratio of 1:2 in a 250 mL volumetric flask.

#### 2.3.4. Determination of Viscosity

The viscosity of the finished product (fusidic acid cream) was determined in triplicate using the Brookfield IV viscometer # (QCA-036): Spindle # 4, 30 rpm, 50 mL beaker at a temperature of 22 °C. The readings were taken after 3 min [33].

#### 2.3.5. Organoleptic Characteristics

All formulations were observed visually for their texture, color, homogeneity, and phase separation. After two minutes of skin application, the feel test was performed. In accordance with the procedure described by Tchienou, the spreadability test was performed on samples in triplicate at zero time, one month, and three months at 25 °C and under accelerated conditions (40 °C/75% 5% RH) [34]. The spreadability of the creams was determined by measuring the spreading diameter of 1 g of the sample placed between two horizontal glass plates measuring 10 cm × 20 cm after one minute
S = m × l/t (S − spreadability test),
where:

m: weight tied on the upper slide, l: length of the glass slide, t: time in s.

#### 2.3.6. Minimal Inhibitory Concentrations for the Selected Novel Topical Cream F2 and Individual Active Substances

The broth microdilution method evaluated the susceptibility of FRSA to the selected formula. It was performed on sterile microtiter plates. The inoculum density was set to 0.5 McFarland, diluted 10-fold in sterile saline, and 5 μL of this suspension was inoculated into 0.1 mL of CAMHB-cation-adjusted Mueller–Hinton broth to reach the final inoculum of 5 × 104 cfu/well. The novel topical cream was diluted in DMSO and added to CAMHB from 2560 μg/mL to 1.25 μg/mL by two-fold dilution in 96-well microtiter plates. After inoculation, plates were incubated at 36 °C for 24 h. The MIC was determined as the lowest antimicrobial agent concentration that inhibits the visible growth of a microorganism in the broth dilution susceptibility test from the wells without visible growth. Ten microliters in CAMHA were cultured and incubated at 36 °C for 24 h [17,35].

Serial dilutions for each formulation were prepared in the diagnostic sensitivity test (DST) for the following microbes: Candida albicans (ATCC # 10231), Escherichia coli (ATCC # 8739), Pseudomonas aeruginosa (ATCC # 9027), Staphylococcus aureus (ATCC # 6538). All dilutions were made with freshly prepared autoclaved agar cooled to a temperature of 47–50 °C in an oven before adding a formulation.

The MIC was determined for the cream formulation by the agar dilution assay against FRSA strains to 1 × 10^5^ CFU to prepare the inoculums. A sample of 0.5 g from the formula in a final volume of 50 mL agar was prepared. Further serial dilutions with agar were performed from this solution to obtain a 0.5–0.0001% concentration. The lowest concentration of formula that prevented bacterial growth is considered the MIC of the formula for all pathogens. The F2 formula and the individual active ingredients (fusidic acid, zinc sulfate, copper sulfate, oleuropein, and thyme oil) were evaluated for the MIC.

#### 2.3.7. Stability Studies

The three formulations (F1, F2, and F3) were stored at accelerated conditions (40 ± 2 °C/75% ± 5% RH). At zero time, one month, and three months, they were evaluated to determine their antibacterial activity and physical stability (appearance and spreadability). The formula with the optimal antibacterial activity and physical stability was selected for further evaluation.

The final optimal formulation (F2) was stored at 25 °C and 40 °C and evaluated after three months for its physicochemical stability. The viscosity of the formula was determined every month over three months. In addition, the formula and FucidinTM were tested for antibacterial activity against Candida albicans (ATCC # 10231), Aspergillus niger (ATCC # 16404), Escherichia coli (ATCC # 8739), Pseudomonas aeruginosa (ATCC # 9027), and Staphylococcus aureus (ATCC # 6538) by the agar well diffusion assay over a three-month period at 25 °C and 40 °C.

## 3. Results

In this study, three different cream formulations were developed (Table 1). All formulations were divided into three aliquots, as shown in Table 2, to evaluate their antibacterial activity and physical stability at accelerated conditions (40 ± 2 °C/75% ± 5% RH). In addition, fusidic acid cream from the local market (Fucidin TM), as a control, was stored under the same conditions and evaluated. The mean (*n* = 3) of the zone of inhibition for microorganism strains for each formulation is shown in Table 2. The in vitro antibacterial activity was determined by measuring the diameter of the zone of inhibition on the agar plate for bacteria, yeast, and FRSA samples. According to the Clinical Laboratory Standards Institute (CLSI) guidelines, the acceptable reported quality control limits for *S. aureus* and *E. coli* are 19–27 mm and 19–26 mm.

The results revealed that formula F1 did not appear to have a desirable antibacterial activity against *E. coli*. No inhibition zones were observed on the agar plates with FRSA samples; therefore, they were discontinued from further evaluation for this formula. The results also showed that trade cream Fucidin TM has limited antimicrobial activity against *S. aureus*; at the same time, it did not have antibacterial activity against *E. coli* and FRSA (Table 2).

The F2 and F3 formulas, which contain higher quantities of oleuropein, zinc sulfate, and copper sulfate than F1, showed desirable antibacterial activity with distinct zones of inhibition observed on the agar plates. Furthermore, the inhibition zones increased as the concentration of oleuropein, zinc sulfate, and copper sulfate increased, indicating concentration-dependent antimicrobial activity for eradicating S. aureus, *E. coli*, and FRSA.

The two cream formulations (F2 and F3) showed antibacterial activity. The F3 formula was shown to have a slightly greater measurement for the diameter of the zone of inhibition despite having higher amounts of oleuropein, zinc sulfate, and copper sulfate than F2. However, the physical appearance of the F3 formula was unacceptable. Formula F3 was bold blue–greenish in nature, the color formed due to increasing the concentration of oleuropein and copper sulfate. Additionally, at 30 rpm, the viscosities for the cream formulations were observed to be 151,601.25 cps for F2, 184,501.05 cps for F3, and the targeted viscosity was to be comparable to fucidin TM’s viscosity (145,501.05 cps). Higher viscosity values might negatively impact the rate of drug released from the formulation vehicle, which would decrease the amount of drug that is available for skin penetration. On the other hand, formula F2 exhibited better spreadability than formula F3. The F2 formula displayed attractive physical characteristics, viscosity, and desirable antibacterial activity; hence, it was selected as the topical cream formula for further studies and preparation for this project.

### 3.1. Physicochemical Properties Evaluations

The physicochemical properties of formula F2 were investigated for three months at 25 °C and 40 °C. The results indicated an excellent appearance and homogeneity for the cream formulation (Table 3). The physical appearance of the cream formulation was light green, matching the natural color obtained from Oleuropine and copper sulfate. The smell of thyme was present in the formulation. In addition, the texture of the cream was smooth with no phase separation. Furthermore, the physicochemical properties of the F2 formula at both conditions remained relatively similar.

### 3.2. Spreadability Test

The spreadability of the cream formulations is directly related to the efficacy of the topical treatment, which predicates the patient spreading the semi-solid formulation in an even layer to manage a standard dose [36].

The values in Figure 1 illustrate the spreading diameter after one minute. The spreadability diameter indicates the range of the area to which the formulation easily expands upon skin application by a small amount of shear [37]. No apparent change in spreadability was observed with increased storage conditions from 25 °C to 40 °C (*n* = 3). The mean ± standard deviation of spreadability for the F2 samples at different conditions ranged from 23.5 ± 0.25 g·cm/s to 23.48 ± 0.3 g·cm/s. The results showed that formula F2 is more likely to have better spreadability than trade products.

### 3.3. HLPC Analysis of Fusidic Acid and Oleuropein

By testing the percentage of fusidic acid and oleuropein by HPLC in the F2 cream, the results showed that the assay of oleuropein was very close in both conditions, 25 °C and 40 °C. Oleuropein in both conditions decreased by about 3% after three months of storage.

A similar result was obtained for fusidic acid after three months of storage at 25 °C and 40 °C, where the assay ranged from 102% at zero days and decreased to 99% and 98% after one and three months of storage, respectively, as shown in Table 4. These results indicate that both fusidic acid and oleuropein are stable and there is no interaction with the other ingredients in the formula of F2.

### 3.4. Determination of Zinc Sulfate and Copper Sulfate

The zinc sulfate and copper sulfate quantities in formulation F2 were analyzed at zero and after three months (Table 5). After achieving the green endpoint, it was noticed that a somewhat low percent of content was obtained from the EDTA titration analysis from zero to the third month, which decreased from 99.8 to 96.2% for zinc sulfate and from 98.7 to 93.4% for copper sulfate at 25 °C. Relatively similar results were obtained at 40 °C, where the assay decreased from 99.3 to 95.4% for zinc sulfate and from 98.5 to 92% for copper sulfate from zero time to the third month. One of the reasons for such a decreased percent of element content might be simple interaction with cetostearyl alcohol with metal salts. However, in this formula, this incompatibility was relatively minor. Furthermore, diluting the sample solution with ethanol in a 1:1 ratio can avoid high temperatures to titrate. The results in this study were better than previous research projects conducted using ICP-MS analysis. A relatively low percentage of metal ions was obtained between 24% for copper and 21% for zinc [27].

### 3.5. Viscosity Measurement

The viscosities of the F2 formula and FucidinTM were determined, and the results are shown in Figure 2. The viscosity for F2 at 30 rpm was found to be 15,160 ± 1.25 cps, while Fucidin TM was found to be 14,550 ± 1.05 cps, which are within the acceptable limits [24]. The viscosity decreased over shear stress, and it was clearly explained that the viscosity of F2 at 0 time and 3 months at 40 °C was found within acceptable limits. It is supposed that a non-Newtonian behavior with the pseudoplastic flow was seen in both formulas. It is evident from Figure 2 that as the rpm or shear stress increased, the viscosity decreased in both formulations. In conclusion, the resistance to flow of FucidinTM was low compared to F2 by pouring the formulations out of the container.

### 3.6. Determination of the MICs for Cream Formulations, Their Active Substances, and FRSA Sample

Fusidic acid is primarily active against gram-positive bacteria such as *Staphylococcus* spp., with no activity against *E. coli*, *P. aeruginous*, and *C. albicans*, the same as the results shown with Fucidin TM (Control), reflecting the growth of the microorganisms tested on agar plates, except for staphylococci. Interestingly, the F2 formula has broad antimicrobial activity in vitro; it was highly active against all microbes by inhibiting the growth of all pathogens as compared with fucidinTM; see Figure 3A–E after storage at accelerated conditions (40 ± 2 °C/75% ± 5% RH) at zero time only.

Parallel with the above test, the F2 formula was evaluated to test its antibacterial activity against FRSA colonies. Twenty-three FRSA isolates were collected from Palestinian hospitals (Beit-Jala Governmental Hospital, Istishari Hospital, Bethlehem Arab Society for Rehabilitation (BASR), and Surgery and Caritas Baby Hospital (CBH)) between January and October 2018. All clinical isolates (FRSA) were susceptible to the F2 formula, with a zone of inhibition greater than 19 mm. One of the FARS isolates (Figure 4A) was obtained from a breast cancer and nodular melanoma patient. It was resistant to ceftriaxone (CT), fusidic acid (FA), gentamycin (GM), ciprofloxacin (CP), and meropenem (MR). However, as shown in Figure 4B, the F2 formula showed antibacterial activity against the FRSA isolate.

There were no zones of inhibition observed on the agar plates with fucidinTM in the FRSA samples (Figure 4A), but the zone of inhibition increased in the F2 (Figure 4B) plate to an average of 37.25 mm from the first to the third month.

After storing the F2 formula for one month at accelerated conditions (40 ± 2 °C/75% ± 5% RH), the MICs of the formula F2 were as presented in Table 6. In addition, the results of the MIC of each active ingredient are shown in Table 6.

Our results demonstrated potent activity against *S. aureus* and *E. coli*. (MIC 10.5 µg/mL for EC (F2) as the entire cream or fusidic acid alone as part of the cream). However, there was also an effect on *C. albicans* and *P. aurginosa* (MIC 85 µg/mL for the entire cream). Furthermore, our additives (oleuropein, thyme oil, copper sulfate, and zinc sulfate) showed moderately effective activity against the pathogen strains with specific MIC values.

### 3.7. Stability of Formula F2

The formulation F2 was stored at 25 °C and under accelerated conditions (40 ± 2 °C/75% ± 5% RH) for three months in plastic containers (*n* = 3). The results showed that F2 was stable after three months under both conditions. There was no precipitation and no change in appearance, color, or pH (Table 7). The fusidic acid, oleuropein, and zinc sulfate assays were within the acceptable range. The assay of copper sulfate was decreased by about 6.5% after three months of storage at both conditions. The formula was evaluated for antibacterial activity after three months of storage. As shown in Table 8, the F2 cream has an antibacterial activity where the observed inhibition zones were above 19 cm.

## 4. Discussion

The emergence of antimicrobial resistance to currently available antibiotics is a significant worldwide problem. Designing new agents or methods to overcome resistance has been performed by either increasing the antimicrobial dosage or combining agents with a different mechanism of action. Metal ions have antimicrobial potential and have been used since long ago; furthermore, metals have a synergistic activity when combined with other antimicrobial agents to overcome multidrug-resistant pathogens, which has been the focus of research currently in treating infections [16,38]. Fusidic acid is a narrow-spectrum bacteriostatic steroid antibiotic commonly used to treat infections caused by MRSA. Resistance to fusidic acid due to prolonged use is mediated by the FusB family of proteins. The FusB protein binds to elongation factor G (EF-G) and promotes dissociation of the antibiotic from its target, restoring normal protein synthesis to proceed uninterrupted [39]. High rates of antibiotic resistance of *S. aureus* to Fusidic acid have been associated with increased usage and prolonged therapy. The prevalence of *S. aureus* resistance to fusidic acid was reported as 10.7% in some studies. The highest resistance to fusidic acid has been reported in Greece at a rate of 62.4% [40]. This study formulated a cream containing metal ions complexed with fusidic acid, oleuropein, and thyme oil with antimicrobial activity against tested microorganisms, including *S. aureus*, *E. coli*, *P. aeruginosa*, *C. albicans*, and *A. niger*.

The formula developed in this study was developed without adding chemical preservatives. Instead, the novel formulation was developed by incorporating natural medicinal products (oleuropein and thyme oil) with significant antibacterial effects [29]. Several investigations found that the EO of thyme has considerable antibacterial action against bacteria and most of the fungal species studied by these authors. In contrast, the extract only had moderate antimicrobial activity against bacteria and was ineffective against fungus [39].

Furthermore, metal ions, copper sulfate, and zinc sulfate, which were studied as antibacterial agents in previous research, had a synergistic antibacterial activity with fusidic acid against *Staphylococcus aureus* and *E. coli* [40]. Other studies showed that using copper to prevent nosocomial infections is promising. It has been reported that the growth of all drug-resistant Gram-negative clinical isolates, including K. pneumoniae and *E. coli*, and *S. aureus* have been significantly inhibited at 1600 µg/mL of copper [41].

In this study, two metal ions were used; copper and zinc showed enhanced antimicrobial activity. In the present research, formula F1 with a low concentration of oleuropein, zinc sulfate, and copper sulfate (Table 2) did not appear to have desirable antibacterial activity against *E. coli*, whereas an increase in the concentrations of these ingredients (F2) had significant antibacterial activity. Furthermore, the results revealed that the inhibition zones increased as the concentration of oleuropein, thyme oil, zinc sulfate, and copper sulfate increased. This showed that the activity of oleuropein, thyme oil, zinc sulfate, and copper sulfate against *S. aureus*, *E.coli*, and FRSA is directly related to their concentrations.

Regarding the preservative effectiveness, formula F2 was observed to have a similar diameter to the zone of inhibition on the agar plate. This work evaluated the lowest concentrations of excipients that could give the best antibacterial effect as a preservative.

From the results in Table 2, the physical appearance of formula F3was unacceptable, which was bold blue–greenish in nature. The color formed due to increasing the concentration of oleuropein and copper sulfate. On the other hand, it was found that formula F2 exhibited better spreadability than formula F3, as shown in Table 2.

The desirable antibacterial activity was demonstrated by formulation F2, making it the appropriate choice to prepare the topical cream for the final formulation. Therefore, further studies are required for formula F2 only. For this purpose, samples from formula F2 were stored at room temperature (25) and under accelerated conditions (40 ± 2 °C/75% ± 5% RH). Furthermore, these samples were evaluated at zero time, one month, and after three months to assess their antibacterial activity and physical and chemical stability.

## 5. Conclusions

The combination of fusidic acid, metal ions, and natural preservatives in a topical cream possesses antibacterial and antifungal properties. The formulated product effectively eradicates multidrug-resistant Gram-negative bacteria, MSSA and FRSA. In addition, it has acceptable physical and chemical properties, stability after prolonged storage for three months at room temperature, and accelerated conditions (40 ± 2 °C/75% ± 5% RH). However, the absence of ex vivo permeation investigations on human stratum corneum was a limitation of the current study. Furthermore, this finding lays the foundation for future studies to uncover these compounds’ pharmacodynamic and kinetics properties.

## Figures and Tables

**Figure 1 pharmaceutics-14-01638-f001:**
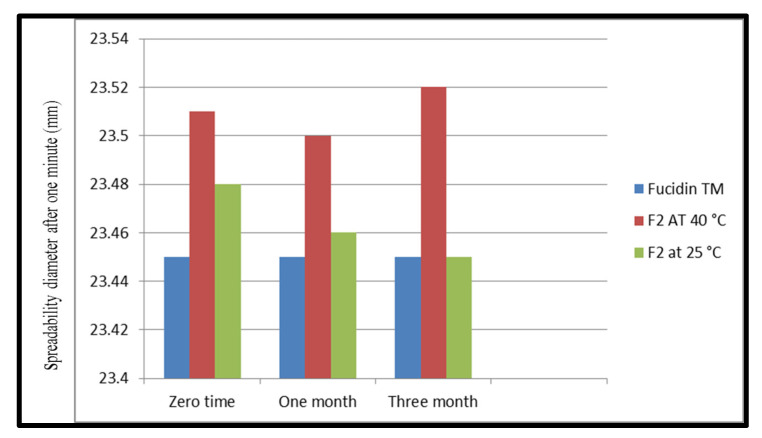
Spreadability values for F2 and fucidinTM at zero time, 1 month, and after 3 months at 25 °C and accelerated conditions (40 ± 2 °C/75% ± 5% RH).

**Figure 2 pharmaceutics-14-01638-f002:**
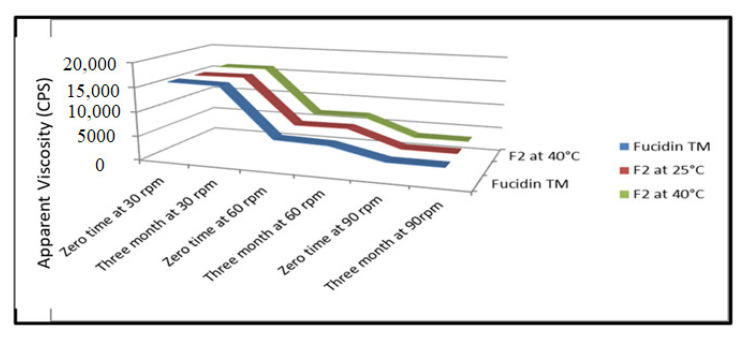
Viscosity of the F2 formula and fucidinTM at zero months and three months at 25 °C and accelerated conditions (40 ± 2 °C/75% ± 5% RH).

**Figure 3 pharmaceutics-14-01638-f003:**
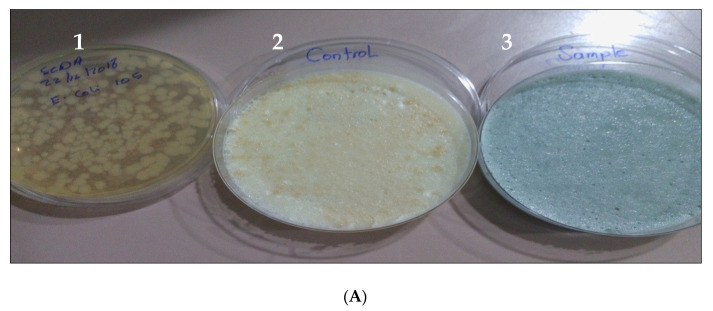
(**A**) (1): Cultured pathogen, (2): FucidinTM (control), and (3): F2 (sample) effect on E.coli isolate. (**B**) (1): Cultured pathogen, (2): FucidinTM (control), and (3): F2 (sample) effect on *C. albicans* isolate. (**C**) (1): Cultured pathogen, (2): FucidinTM (control), and (3): F2 (sample) effect on *A. niger* isolate. (**D**) (1): Cultured pathogen, (2): FucidinTM (control), and (3): F2 (sample) effect on *P. aeruginosa* isolate. (**E**) (1): Cultured pathogen, (2): Fucidin TM (control) and (3): F2 (sample) effect on *S. aureus* isolate.

**Figure 4 pharmaceutics-14-01638-f004:**
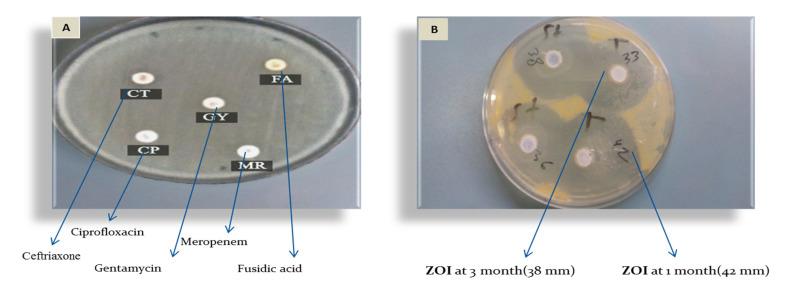
Antimicrobial susceptibility testing of *S. aureus* by disk diffusion method. (**A**): FRSA plate shows the resistance effect of different antimicrobial agents on *S. aureus* isolate to CT: ceftriaxone, FA: fusidic acid, GY: gentamycin, CP: ciprofloxacin, MR: meropenem. (**B**): FRSA plate F2 formula sensitive *S. aureus* shows the zone of inhibition more than 19 mm, T: at zero time, st: at the third month.

**Table 1 pharmaceutics-14-01638-t001:** Qualitative and quantitative compositions of 3 fusidic acid cream formulations.

No.	Component	F1/g	F2/g	F3/g	Function
1	Fusidic acid	2	2	2	Active ingredient (API)
2	Zinc sulfate	1	1	2	Antibacterial, antiviral
3	Copper sulfate	1	1.5	2	Antibacterial, antifungal
4	Oleuropein	0.2	0.4	0.6	Antioxidant and preservative
5	Thyme oil	0.1 mL	0.1 mL	0.1 mL	Antioxidant and preservative
6	Cetosteroyl alcohol	7	7	7	Emollient
7	Macrogol A6	1.5	1.5	1.5	Emulsifying agent
8	Macrogol A25	1.5	1.5	1.5	Emulsifying agent
9	Paraffin oil	12	12	12	Vehicle
10	Propylene glycol	8	8	8	Solvent
11	Purified water	65.7	64.5	63.3	Vehicle

**Table 2 pharmaceutics-14-01638-t002:** Evaluation of different topical formulations at 0, 1, and 3 months at accelerated conditions (40 ± 2 °C/75% ± 5% RH).

Characteristic /Sample	Zone of Inhibition (mm)								
	F1			F2			F3		
	*S. aureus*	*E.coli*	FRSA	*S. aureus*	*E. coli*	FRSA	*S. aureus*	*E. coli*	FRSA
Control (Fucidin TM)	18.3	0	0	18.3	0	0	18.3	0	0
Aliquot Test (1) zero time	22.9	14.5	15.4	39.1	21.6	36.7	39.6	24.6	38.8
Aliquot Test (2) one month	22.4	14.3	15.2	36.8	21.4	34.5	38.2	23.8	37.6
Aliquot Test (3) three months	21.2	12.6	14.1	35.6	20.3	33.8	37.4	22.7	36.4
Appearance	Light green			Light green			Bold blue–greenish		
Spreadability	Good spreadability			Good spreadability			Thick spreadability		

**Table 3 pharmaceutics-14-01638-t003:** Physicochemical evaluations for the F2 formula at zero time, one month, and after three months at 25 °C and accelerated conditions (40 ± 2 °C/75% ± 5% RH).

Time	Odor	Physical Appearance	Feel after Application	Texture	Color	Phase Separation	Homogeneity
Zero months	Light smell of thyme	Opaque	Moisture	Smooth	Light green	No	Homogenous
One month	Light smell of thyme	Opaque	Moisture	Smooth	Light green	No	Homogenous
Three months	Light smell of thyme	Opaque	Moisture	Smooth	Light green	No	Homogenous

**Table 4 pharmaceutics-14-01638-t004:** Percent fusidic acid and oleuropein content in formula F2 at zero time, one month, and three months at 25 °C and accelerated conditions (40 ± 2 °C/75% ± 5% RH) (*n* = 3).

Assay of Agents at Different Conditions	Zero Time	One Month	Three Months
Fusidic acid at 25 °C	102 ± 0.11	99 ± 0.13	98 ± 0.13
Fusidic acid at 40 °C	101.4 ± 0.12	98.9 ± 0.10	97.8 ± 0.12
Oleuropin at 25 °C	98.6 ± 0.24	97.3 ± 0.26	95 ± 0.28
Oleuropin at 40 °C	98 ± 0.37	97 ± 0.42	95 ± 0.43

**Table 5 pharmaceutics-14-01638-t005:** Percent zinc sulfate and copper sulfate content in formula F2 at zero time and 3 months at 25 °C and accelerated conditions (40 ± 2 °C/75% ± 5% RH) (*n* = 3).

Assay of Agents under Different Conditions	Zero Time	Three Months
Zinc Sulfate at 25 °C	99.8 ± 0.82	96.2 ± 0.48
Zinc Sulfate at 40 °C	99.3 ± 0.73	95.4 ± 0.53
Copper Sulfate at 25 °C	98.7 ± 1.23	93.4 ± 1.60
Copper Sulfate at 40 °C	98.5 ± 1.28	92 ± 1.53

**Table 6 pharmaceutics-14-01638-t006:** MICs of the F2 formula and active ingredients alone against the selected pathogens.

	* C. albicans *	* E. coli *	* S. aureus *	* P. aeruginosa *
**F2**	**85**	**14**	**10.5**	**150**
Fusidic acid	1.7	0.28	0.21	3
Copper sulfate	1.3	0.21	0.16	2.25
Zinc sulfate	1.3	0.21	0.16	2.25
Oleuropin	0.3	0.06	0.04	0.6
Thyme oil	0.1	0.01	0.01	0.15

**Table 7 pharmaceutics-14-01638-t007:** Physical parameters and assay results (as percentage) of formula condition at zero time and 3 months at 25 °C and accelerated condition (40 ± 2 °C/75% ± 5% RH) (*n* = 3).

Comparisons	Time	Precipitation	Appearance	Color	PH	Assay of Zinc Sulfate	Assay of Copper Sulfate	Assay of Oleuropine	Assay of
									Fusidic acid
Fusidic acid at 40 °C	0 time	Negative	Uniform	Light green	4.21 ± 0.02	99.3 ± 0.73	98.5 ± 1.25	98 ± 0.37	101.4 ± 0.12
	1 month	Negative	Uniform	Light green	4.19 ± 0.04	96.8 ± 0.59	98.3 ± 1.21	97 ± 0.42	98.9 ± 0.10
	3 months	Negative	Uniform	Light green	4.16 ± 0.03	95.4 ± 0.53	92 ± 1.53	95 ± 0.43	97.8 ± 0.12
Fusidic acid at 25 °C	0 time	Negative	Uniform	Light green	4.23 ± 0.05	99.8 ± 0.82	98.7 ± 1.23	98.6 ± 0.42	102 ± 0.11
	1 month	Negative	Uniform	Light green	4.22 ± 0.04	99.6 ± 0.79	96.4 ± 1.05	97.3 ± 0.26	99 ± 0.13

**Table 8 pharmaceutics-14-01638-t008:** Antibacterial activity of the F2 formulation over a three-month period at 4 °C, 25 °C, and 40 °C (*n* = 3).

	Period	Zone of Inhibition (mm)		
		Temperature (°C)		
		4 °C	25 °C	40 °C
Plastic container	Zero time	-	25.35 ± 0.21	24.8 ± 1.41
	One month	-	24.55 ± 0.28	24.1 ± 1.10
	Two months	-	23.85 ± 0.22	-
	Three months	22.8 ± 0.07	22.38 ± 0.24	24.0 ± 1.30

## Data Availability

The data used to aid the outputs of this research are available from Hani Naseef (hshtaya@birzeit.edu) upon request.

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
