# Peer review of "Novel Fusidic Acid Cream Containing Metal Ions and Natural Products against Multidrug-Resistant Bacteria"

_pharmaceutics, 2022, doi:10.3390/pharmaceutics14081638_

Round 1
Reviewer 1 Report
The present study was focused on a simple strategy to improve the antibacterial activity of fusidic acid topically applied as creams by combining it with two metal ions and two natural products (oleuropein and thyme oil) presenting synergistic effect. This study contributes to the enlargement of data base for dermatologic semisolid preparations effective against fusidic acid-resistant Staphylococcus aureus strains. The title and abstract are appropriate for the content of the text. The article is well constructed and clearly written. Also, the experiments were well conducted by a generally sound methodology used for the characterization of the studied formulation. The methods are clearly described, and the results are supported by enough references and statistical analysis.
For these reasons, I recommend the publication of this work. However, the authors should clarify the following two items:
- In the 4th paragraph of the Results section: The authors should explain more detailed the reasons for considering formula F3 unacceptable, since this formulation showed the highest antibacterial activity; why the intense color and the “thick spreadability” were considered the exclusion criteria for this formulation?
- In Conclusions section. It would be best that the authors state the limitations of their study.
Also, the authors should correct the typos throughout all the manuscript.
Author Response
Dear reviewer
I appreciate your input. We carefully evaluated the valuable comments and adjusted the manuscript as necessary. Our point-by-point responses and the related adjustments are attached.
Please be aware that the Changes Outline report addresses the page and line numbers when the manuscript's track changes feature is turned on.

Reviewer 2 Report
In this paper, the authors investigated that effect of Fusidic acid cream with metal ions and natural products on antibacterial activity. This is an interesting result to overcome multidrug-resistant bacteria. However, several points needed to be addressed by authors to improve the quality of the paper. Therefore, this paper can be accepted after revision of this paper considering the comments below.
1. In this study, all formulations the authors developed have Fusidic acid and multiple metal ions and natural products. The authors mentioned that synergistic effect, not addictive effect of Fusidic acid with metal ions and natural products was evaluated in this study. However, the role of each component for synergistic effect for activity of Fusidic acid is unclear. How do these components contribute to activity of Fusidic acid?
2. F2 and F3 formulations showed antibacterial activity against S. aureus, E.coli, FRSA. The authors mentioned that these formulations indicated concentration-dependent activity. However, I think F2 effect is virtually the same as F3. Please explain more about this result.
3. How did authors evaluate spreadability of formulations in Figure 1? There is no explanation of detail in methods section. I strongly recommend that you use spreadmeter in Spreadability test to get accurate data.
4. There are inappropriate write in many parts of the manuscript. For example, unnecessary hyphens (line 497, activi-ty, etc.) and wrong font of “°” (line 26, oC, etc.) and more are found. The authors should properly correct that in the whole manuscript.
5. The vertical axis in Figure 2 is unclear. I recommend it be modified to be clear.
I hope these comments will be helpful for the improvement of the paper.
Author Response

(The authors gave the same response as above.)

Round 2
Reviewer 2 Report
Dear authors,
The revised manuscript has been improved well. I recommend publication with only a modest amount of revision. I require additional explaination on following point about comment 1.
The authors mentioned that Fusidic acid, which inhibits protein synthesis and is considered bacteriostatic when used alone, has no activity against (FSRSA), but the combination product has excellent activity.
The all formulations studied in this paper include Fusidic acis. When the formulation without Fusidic acid, that is, only metal ion and natural products, does it show low antibiotic activtity? The aouthers need to explain that the activity of the formulation is not only due to metal ion and natural products, to support your claim of synergistic activity.
Author Response
Actually, we tested Fucidin alone and in combination with metal ions and natural products. The metal ion and natural products was not tested alone without Fucidin, this may be considered as a limitation of the study. However, we can see that increasing the concentrations of copper sulfate and Oleuropein in the formulas correlated with increasing zones of inhibition. In addition, the MICs in Table 6, showed that testing the metal ions and natural products alone was less effective (higher MICs g/mL) as compared with Formula F2 which showed much lower MICs in the range of 10.5 mg/ml for S. aureus to 150 mg/ml for P. aeruginosa

Round 3
Reviewer 2 Report
Dear authors,
The manuscript has been revised well.